



# On the interaction of stochastic forcing and regime dynamics

Josh Dorrington[1,2] and Tim Palmer[1]

[1]Atmospheric, Oceanic, and Planetary Physics, University of Oxford, Oxford, UK
[2]Institute for Meteorology and Climate (IMK-TRO), Karlsruher Institute of Technology, Germany

**Correspondence:** Josh Dorrington (joshua.dorrington@kit.edu)

**Abstract.**

In this paper we investigate the curious ability of stochastic forcing to increase the persistence of regimes, in a low-order, stochastically forced system. In recent years, evidence from both simple models and climate simulations have suggested that stochastic forcing can act as a stabilising force to increase regime persistence, but the mechanisms driving this potential re-
inforcement are unclear. Using a six-mode truncation of a barotropic $\beta$-plane model, featuring transitions between analogues of zonal and blocked flow conditions, we show that moderate levels of fast-varying stochastic forcing can increase the low-frequency variability of the system, and act asymmetrically to increase the persistence of certain regimes. We show that the presence of a deterministically-inaccessible unstable fixed point, and the low-dimensionality of the flow during blocking, are vital dynamical components that allow this stochastic persistence to occur. We present a simple geometric argument that ex-
plains how stochastic forcing can slow the growth of instabilities, which may have more general applicability in understanding stochastic chaotic systems.

## 1 Introduction

The idea that the atmospheric flow might possess multiple large-scale quasi-equilibrium states – commonly termed circulation regimes – was initially put forward by Rossby (1940) to describe the empirically observed tendency for European weather
patterns to take on a finite range of frequently recurrent configurations, and was first explored in mathematical detail by Charney and DeVore (1979) (hereafter CdV79). In a low-order barotropic channel model of the mid-latitudes, CdV79 showed that the flow could equilibrate to one of two stable states: an approximately zonally symmetric flow or a stationary wave pattern that provides a low-dimensional analogue of blocking. They suggested that the real world equivalents of these equilibria states might be destabilised by perturbations from unresolved modes ( in this idealised model, this included spatial scales of $\sim \mathcal{O}(1000)$
km), causing unpredictable shifts from one state to another and introducing a vacillation between large-scale weather patterns.

While similar results were obtained in a variety of spectrally truncated models (Charney and Straus, 1980; Kallen, 1981; De Swart, 1988), it was soon realised that these truncated equilibria were often absent or unstable in more detailed models, and were frequently rendered 'unrealisable' by shifts in the model attractor (Reinhold and Pierrehumbert, 1982; Cehelsky and Tung, 1987). While the reproduction of regime dynamics in barotropic annulus experiments (Weeks et al., 1997) suggested that
the dynamics of fixed point regimes could be found even in untruncated flows, a more sophisticated mathematical perspective was needed.



Moving away from stable fixed point dynamics, Itoh and Kimoto (1996) and Itoh and Kimoto (1997) linked regime dynamics to chaotic itinerancy, associating regimes with the merging of different attractors through an explosive bifurcation. The importance of preferred transitions between regimes (Kimoto and Ghil, 1993; Kondrashov et al., 2004; Selten and Branstator, 2004), associated with higher dimensional invariant measures such as homoclinic orbits was also realised.

Despite these, and other, theoretical advances, there are still many open questions about the behaviour of regimes that are challenging to tackle even in simple models, and it can in any case be difficult to map lessons learned from systems with only a handful of dimensions to full earth-system models. One such challenging question is the impact of stochastic forcing on regime systems. Contrary to the suggestion of CdV79, and to linear intuition, there is evidence that fast varying random processes can stabilise regimes rather than destabilise them. Stochastic forcing has been shown to improve regime predictability in truncations of the Lorenz '96 model (Düben et al., 2014; Christensen et al., 2015) and climate model regime representation (Dawson and Palmer, 2015), as well as the representation of other slow-varying climate modes (Palmer and Weisheimer, 2011; Berner et al., 2012). Further, Kwasniok (2014) (hereafter K14) showed directly that additive noise can increase the lifetime of regimes in both the Lorenz '63 system and a chaotic reformulation of the CdV79 model found by Crommelin et al. (2004) (hereafter C04).

The reason for this stochastic persistence is unclear, and in this paper we make a study of the stochastically forced CdV79 model, as retuned by C04, in order to shed light on this question. The retuned system features chaotic dynamics with multi-modal regime behaviour, vacillating between a long-lived blocking state and a more zonal flow state, analogous to the fixed points of CdV79, and without imposition of external forcing or requiring unresolved small-scale instabilities. We introduce the model and its regime structure in section 2. We then discuss the impact of stochasticity on that regime structure in section 3 and propose that increased persistence can be qualitatively understood as a result of the theory of hitting time problems in combination with a simple geometric mechanism whereby stochasticity slows the growth of instabilities. We summarise our results and discuss implications in section 4.

## 2 Model formulation

The deterministic Charney deVore model is obtained by spectral truncation of a barotropic flow in a $\beta$-plane channel geometry. It can be expressed as a six-equation system of ordinary differential equations, derived in full in Appendix A:

$$
\begin{aligned}
\dot{x}_1 &= -C(x_1 - x_1^*) + \tilde{\gamma}_1 x_3 \\
\dot{x}_2 &= -Cx_2 + \beta_1 x_3 - \alpha_1 x_1 x_3 - \delta_1 x_4 x_6 \\
\dot{x}_3 &= -Cx_3 - \beta_1 x_2 + \alpha_1 x_1 x_2 + \delta_1 x_4 x_5 - \gamma_1 x_1 \\
\dot{x}_4 &= -C(x_4 - x_4^*) + \epsilon(x_2 x_6 - x_3 x_5) + \tilde{\gamma}_2 x_6 \\
\dot{x}_5 &= -Cx_5 + \beta_2 x_6 - \alpha_2 x_1 x_6 - \delta_2 x_4 x_3 \\
\dot{x}_6 &= -Cx_6 - \beta_2 x_5 + \alpha_2 x_1 x_5 + \delta_2 x_4 x_2 - \gamma_2 x_4
\end{aligned}
\tag{1}
$$



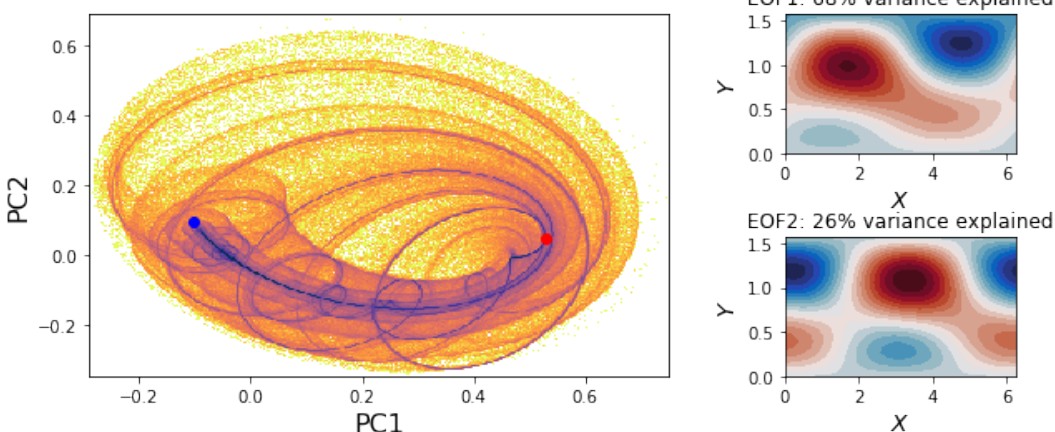

**Figure 1.** Left: Projection of the deterministic attractor into the space of the leading EOFs, capturing 92% of the total variance. Right: The flow patterns corresponding to the first and second EOFs of the system. Blue and red dots show the positions of fixed points corresponding to blocked and zonal flows respectively.

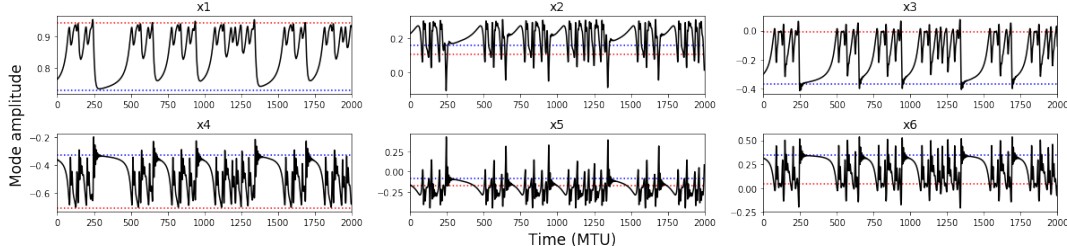

**Figure 2.** A 2000 MTU integration of the CdV79 system, showing the evolution of each mode amplitude. The dynamics are chaotic but weakly so, showing clear quasiperiodic behaviour. Two regimes of behaviour can be seen; fast chaotic oscillations, divided by periods of slow evolution with irregular duration. The mode values at the blocked and zonal fixed points are shown with dashed blue and red lines.

In brief, the six variables $\{x_i\}$ represent spectral mode amplitudes of a streamfunction field, subject to a linear relaxation to a zonally symmetric background state, Coriolis and orographic forces, and quadratic advective nonlinearities. To explore stochastic effects, we introduce an additive white noise vector to the tendency, given by a normal distribution $\xi_i$ with standard deviation $\sigma$, as in K14:

$$\dot{x}_i = \dot{x}_{i,\text{Deterministic}} + \sigma\xi_i \tag{2}$$

The system is integrated using the Euler-Maruyama difference scheme with a timestep of dt=2e-4 model time units (MTU), and is sampled every 1 MTU. We compute integrations of length 200,000 MTU for a range of $\sigma$ values.



To visualise the model phase space, we project the six-dimensional attractor into the space of the two leading empirical

orthogonal functions (EOFs) of the deterministic system, as shown in figure 1, capturing 94% of the system's variability (figure S1 shows the various projections of the six $x$-modes). The attractor is clearly highly structured, with a number of strongly preferred trajectories through phase space (note the logarithmic colour scheme in figure 1). This low dimensional attractor shows no obviously separable phase space regions such as the twin lobes of the Lorenz '63 system, and we see that the trajectories are highly twisted together, producing a scroll-like geometry.

A representative sample of the model's temporal evolution is shown in figure 2. Multimodality is readily apparent, with predictable slowly evolving periods, with near-zero phase speed, corresponding to persistent 'blocked' flow patterns, divided by highly chaotic, fast evolving flows that contain both zonally symmetric states and transient wave activity. The regime structure is therefore quite asymmetrical, in contrast to the Lorenz '63 or '96 systems whose regimes are essentially equivalent to eachother.

Instabilities in the CdV system arise from two basic mechanisms: an orographically-induced saddle-node bifurcation (a transition from one to multiple fixed-point equilibria) as identified in the original CdV79 paper, and a barotropic Hopf bifurcation (a transition from a fixed point to a periodic orbit) associated with the onset of a zonally propagating waves in the model domain. Both bifurcations are codimension 1, existing along a surface with one less dimension than the parameter space, and so can in general be found by tuning 1 model parameter. Stable steady state and periodic wave solutions to equation 1 are common

for arbitrary sets of model parameters (not shown), while chaotic – and especially multimodally chaotic – solutions are harder to find. In the original formulation of CdV79, two stable fixed points were found, corresponding to a zonally symmetric state, and a 'blocked' flow, with strong wave activity.

The model parameters used here, and in C04 and K14, are close to a parameter set where the barotropic Hopf and orographic saddle-node bifurcations merge: one fixed point bifurcates into two fixed points and a limit cycle. This fold-Hopf point has

codimension 2, meaning two model parameters must be tuned to obtain it. High codimension bifurcations seem to be of reduced relevance to the real world, representing exceptional states in parameter space. However, the fold-Hopf bifurcation is known to produce a range of lower codimension, and so more relevant, bifurcations at nearby parameter values, including paths to chaotic dynamics (Champneys and Kirk, 2004). Most relevant to us is the generation of homoclinic orbits, a type of bifurcation where orbits stretch from a fixed point back to itself, with an infinite period. These orbits can lead to fully fledged

chaotic dynamics and the creation of a strange attractor through intersection with an unstable toroidal limit cycle. This creates a 'whirlpool repeller' (Shilnikov et al., 1995) and exactly the scrolling structure seen in figure 1. The two fixed points, analogs of those in CdV79, remain but are now unstable, and at the parameter values chosen here the strange attractor moves between the vicinity of these two fixed points, as seen in figures 1 and 2, shadowing collapsed homoclinic orbits. The flow patterns corresponding to these fixed points are shown in figure S2. See De Swart (1988) and C04 for more details on the bifurcation

structure of the model.

The upshot of this particular route to chaos is that structure of the model attractor and the regime variability of the CdV system is rooted in the dynamics of unstable homoclinic orbits, with close approaches to the unstable fixed point leading to periods of slow evolution, and then rapid 'bursting' to chaotic, more turbulent flow. Such behaviour is seen in other simple at-



mospheric models (Crommelin, 2002) including the Lorenz 84 model (Shilnikov et al., 1995), and in the dynamics of nonlinear
circuits (Pusuluri and Shilnikov, 2018) and neurons (Izhikevich, 2006).

We should emphasise that this type of regime behaviour is of a fundamentally different origin and type to that of the archetypal Lorenz '63 system. In Lorenz '63, crisis-induced intermittency creates regime dynamics when the basin of attraction of two attractors merge together, triggering movement between two distinct regimes of fully chaotic behaviour. The Charney deVore system, in contrast, experiences almost deterministic dynamics before drifting away and entering a transient chaotic regime. At an unpredictable later time, the system will once again return to the neighbourhood of the periodic orbit, and reenter the slow-evolving regime. For a recent discussion of the co-existence of predictable and chaotic flow states in atmospheric systems see Shen et al. (2021).

## 2.1 Regimes in CdV

When studying a multimodal system, bulk metrics such as the mean or variance can obscure the underlying dynamical structure. Therefore it is natural to use a regime framework as a way of partitioning phase space into dynamically distinct regions of interest. Following the approach of K14 we use a hidden Markov model (HMM), to identify qualitatively different dynamical regimes, with the minor methodological change of fitting the HMM to the full six-dimensional state vector, rather than restricting to the $x_1$-$x_4$ subspace as in K14. As for most clustering methods, we must make an *a priori* choice of regime number when fitting the HMM. We use three regimes, which were found in K14 to well separate the qualitatively distinct model dynamics, capturing both the persistent quasi-predictable blocking regime, the highly chaotic zonal regime, and the transitions between them. Figure 3a) shows the regions of the deterministic attractor assigned to each regime, and figure S3 shows the corresponding flow patterns. The blocking regime is small in phase space extent, and is focused on the tightly spiralled trajectories that start close to the blocking fixed point. It clearly separates from the swirling trajectories assigned to the transitional regime. The zonal state contains the very chaotic 'branching' of the attractor into different orbits, and contains the early parts of these orbits until their transition back to blocking becomes unavoidable. Interestingly, these regimes broadly correspond to regions of the attractor with different numbers of unstable dimensions (figure 3b)) supporting our choice of 3 regimes as dynamically appropriate. Our regime assignment also has some similarities to those found in Strommen et al. (2021), which used persistent homology to identify regimes in the CdV system and found a persistent component similar to our blocking state.

For $\sigma \neq 0$, we identify regimes in the same way, refitting the HMM for each $\sigma$ value. This is necessary as while the qualitative shape of the attractor[1] is robust over a range of noise amplitudes, the fine structure seen in figure 1 breaks down and the stochastic simulations experience shifts of probability density closer to the location of the blocking fixed point (shown in figure S4).

Figure 4 shows the temporal evolution of the zonally symmetric $x_1$ mode, and the phase velocity $v_t = \frac{1}{2}\sqrt{\sum_{i=1}^{6}(x_{i,t+1} - x_{i,t-1})^2}$, for different $\sigma$ values, coloured by regime assignment. The characteristics of the regimes are broadly preserved even up to $\sigma$=1.6e-2: the persistent blocking state has low $x_1$ values and very low phase velocities, the zonal state has high $x_1$ values

---

[1]While it is not formally accurate to speak of a model attractor for a stochastic system, we shall continue to do so as a shorthand for the well-converged probability density function of the model.




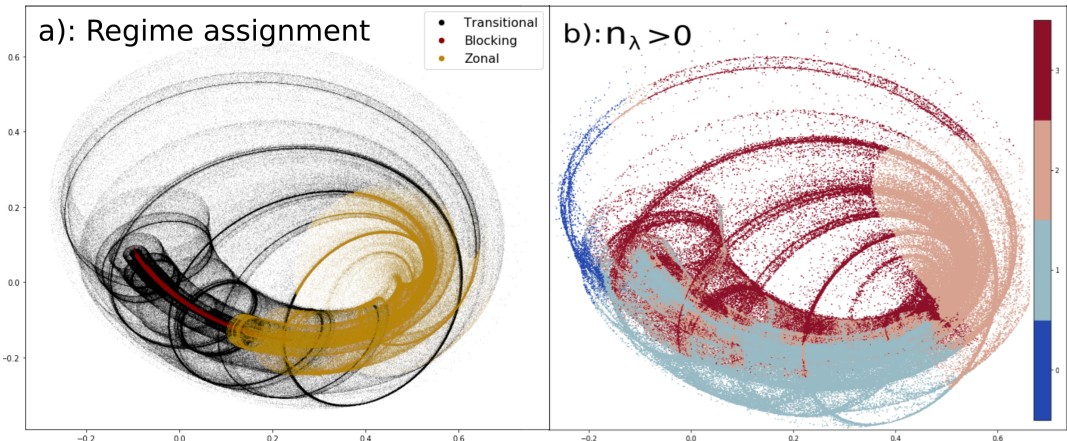

**Figure 3.** a) The assignment of points on the deterministic model attractor to the three hidden Markov model regimes projected into the EOF1-EOF2 plane. b) the number of unstable dimensions as identified by the positive local Lyapunov exponents, calculated at each point along the deterministic model attractor, projected into the EOF1-EOF2 plane.

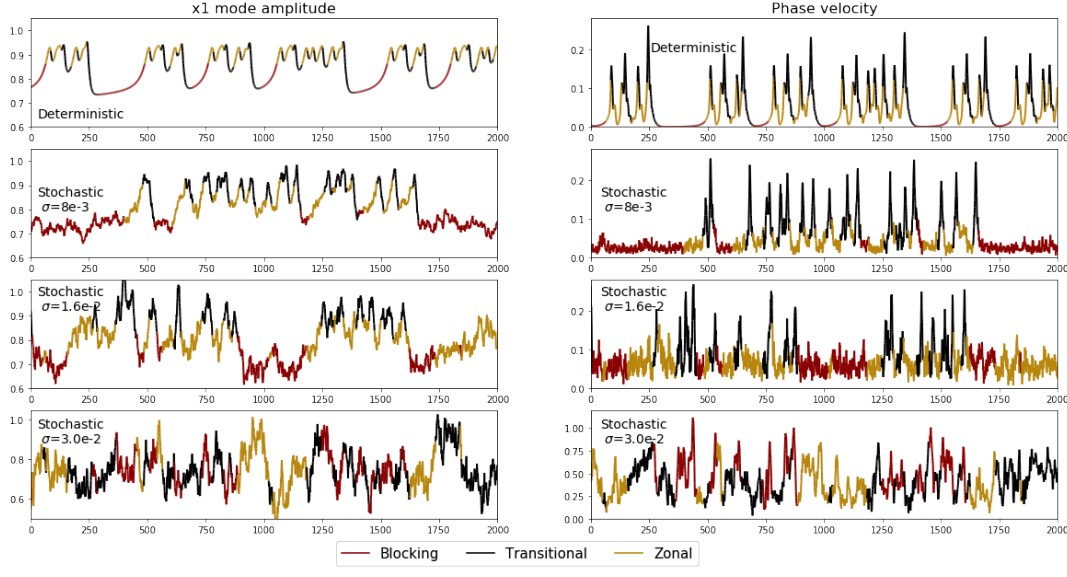

**Figure 4.** Left: Time evolution of the $x_1$ mode in deterministic and stochastic cases, with points coloured by their regime assignment. Right: The corresponding phase velocities.





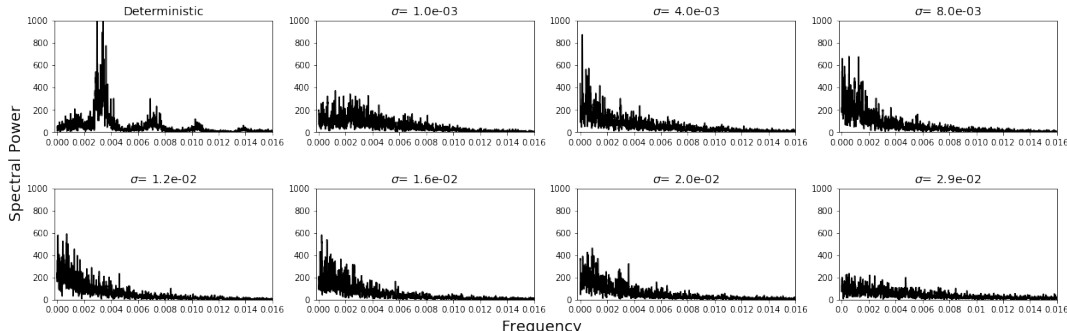

**Figure 5.** The power spectral density of the $x_1$ mode, for a range of stochastic forcing amplitudes, calculated from 200,000 MTU integrations.

and relatively low phase velocities, while the transitional regime has high $x_1$ values and high phase velocity. However, by the time $\sigma$=3.0e-2, it is apparent that there is no meaningful regime structure remaining: the phase velocity never drops below 0.2, indicating a loss of quasistationarity: at this point the deterministic dynamics play second fiddle to the stochastic term, and the assignment of regimes becomes meaningless.

## 3 Stochastic persistence

Figure 5 shows Fourier transforms of the $x_1$ mode for a range of $\sigma$ values. The deterministic spectrum possesses a number of clear peaks, consistent with the quasiperiodic nature of the dynamics, with each peak loosely associated to a preferred trajectory through phase space. The application of even weak stochastic forcing destroys this quasiperiodic structure, producing a much flatter spectra. However, as stochastic forcing increases up to $\sigma = 0.008$, increased power appears in the spectra at low frequencies – far more than in the deterministic case. This indicates that stochasticity is generating new, long-time scale behaviour in the system. As noise amplitude increases beyond this threshold, the spectrum flattens again, eventually reaching a flat white noise spectrum as the stochastic term completely overshadows the deterministic dynamics. Spectra are completely comparable for the other five modes (not shown).

This is in excellent agreement with K14, which showed increasing regime persistence with noise amplitude that peaked at $\sigma = 0.008$ and then decreased. We show here that this is associated with a genuine slowing of the continuous modes visible in the Fourier spectrum, and is not simply an artefact of the regime approach.

With this reassurance, and given that the HMM maintains the same qualitative characteristics for $\sigma$ values $\leq 0.016$, we judge that the regime assignment provides a sensible way to analyse the amplified low frequency variability seen in figure 5 in more detail. Figure 6, shows the distribution of lifetimes for each of the three regimes, as a function of increasing $\sigma$. While all regimes show some increase in regime persistence for moderate noise amplitudes, the effect is by far the most significant in the blocking regime, showing a near trebling of mean regime lifetime that peaks at $\sigma = 0.008$. As shown in figure S5, this peak is associated with a 150% increase in the number of blocks lasting at least 100 MTU, and a 600% increase in those lasting longer than 175 MTU. Changes in the median blocking lifetime however are smaller, and the distribution is clearly skewed,





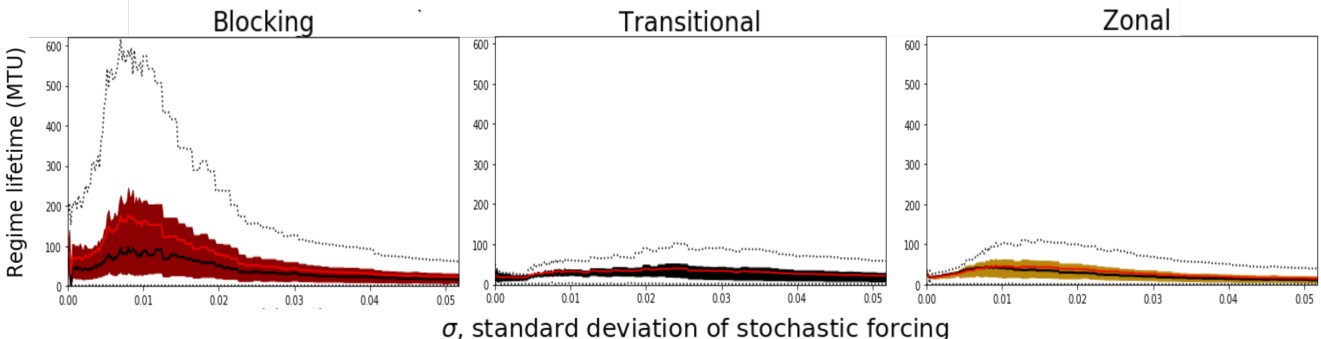

**Figure 6.** The distribution of blocking regime lifetimes, as a function of stochastic forcing amplitude. Shaded regions mark the interquartile range, with the thick black line showing the median, and the red line the mean. Dotted black lines show the 0th and 95th percentiles of the distribution.

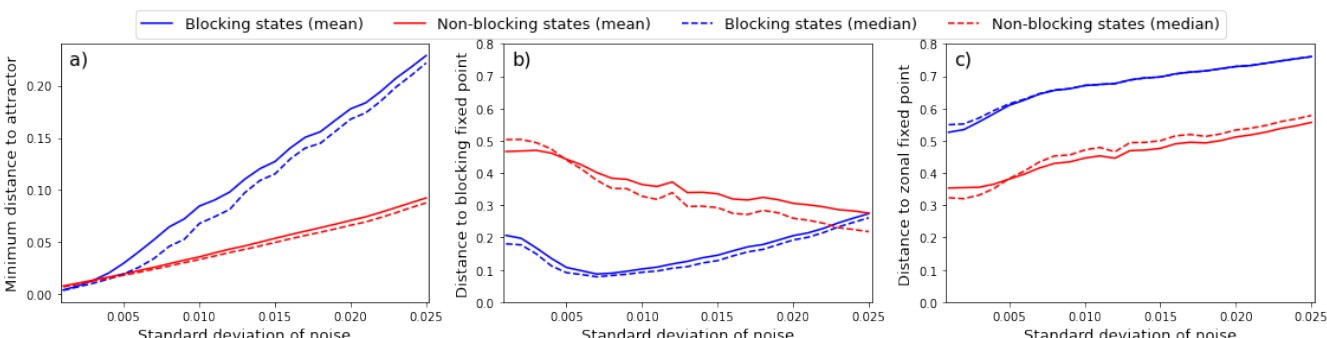

**Figure 7.** a) The mean (solid lines) and median (dashed lines) euclidean distance between points in the stochastic simulation and their closest neighbour on the deterministic attractor, shown for the blocking and non-blocking regimes. b,c) As in a) but showing the distance between points in each simulation and the blocking (b)) and zonal (c)) equilibria respectively.

with the 95th percentile of blocking lifetime exceeding 600 MTU when persistence is maximum. While the other regimes also
show a general increased variance in lifetime as we might expect from a stochastically forced process, the generation of a very
fat-tailed distribution is unique to the blocking regime.

How can we explain the ability of random forcing to stabilise regimes? And why does it act here asymmetrically; primarily
in the blocking regime and less so in the others? One piece of the picture comes from considering mean state changes in the
system. Figure 7a) shows that the average distance between points in the stochastic simulations and their closest neighbour
on the deterministic attractor increases, approximately linearly, as the amplitude of the stochastic forcing increases. Once
again there is asymmetry between regimes, with the minimum distance during blocking growing far faster than during the
other regimes. We see in figures 7b) and c) that this is associated with a drift towards the neighbourhood of the unstable
blocking fixed point, with no equivalent result seen for the zonal fixed point (this can also be seen in figure S4). If we consider





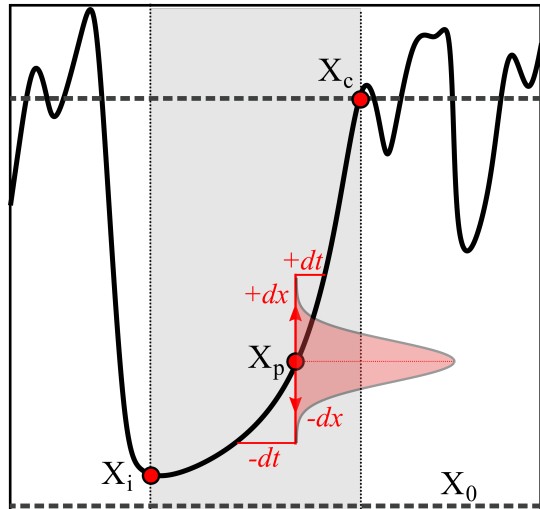

**Figure 8.** A schematic showing one variable, $x$, of a chaotic process, which moves between periods of chaotic variability and periods of predictable, monotonic growth (shaded in grey). The predictable regime is entered at a random position $x_i$, in the neighbourhood of an unstable fixed point $x_0$, and once it reaches an amplitude $x_c$ it will re-enter the chaotic regime. At a position $x_p$, we consider the impact of applying a random perturbation $dx$ to the $x$ variable, drawn from some symmetric probability distribution, with vanishing support outside the interval $[X_0 - X_p, X_c - X_p]$. We see that, due to the convex growth of $x$, a perturbation with negative sign will increase the duration of the predictable period more than a corresponding perturbation with positive sign will shorten it.

the dynamics in the neighbourhood of the blocking fixed point, we realise the deterministic tendency there is near-zero, and
so the additive stochastic terms dominate entirely: the dynamics approach six-dimensional Brownian motion. How long, in expectation, will it take for this Brownian motion to move the system away from the fixed point once again, and return it to a dynamics-dominated flow state? This question is simply a reformulation of the classic hitting time problem: at what time will a random process first reach a distance $a$ from its initial condition? For Brownian motion it is known that in fact the average time taken is infinite: the distribution of hitting times is fat-tailed and so the mean diverges (Kallenberg, 2017). Therefore the skewed
distribution of blocking lifetimes seen in figure 6 can be understood in terms of the amount of time spent near the fixed point, as shown in 7b). As $\sigma$ increases past 0.008, the average distance from both the attractor *and* the blocking fixed point increases. As the stochastic becomes stronger than the deterministic damping along the stable manifold, the system increasingly explores the full six dimensional phase space, and so the directionality of the forcing vanishes.

However, the increased persistence is not entirely a result of excursions to the noise-dominated neighbourhood of the fixed
point. There is also an important interaction between stochasticty and dynamics that derives from the low-dimensional, ordered flow that characterises the blocking regime. Consider a one dimensional system, as in the schematic figure 8, moving between a highly chaotic state and a slowly evolving, predictable state. We could think of this as a parameterisation along the 1D unstable manifold of our CdV79 system. We consider that immediately upon entering the regime of slow evolving dynamics,



the system is at a point $x = x_i$ and will stay in the quasi-stationary regime until it reaches a value $x = x_c$. We assume that $x$

grows monotonically with the time $t$ in this regime, so we can parameterise $x$ in terms of $t$ or vice versa: $f : x \rightarrow t$, $g : t \rightarrow x$.

If we let $x = f(t)$ be any convex, monotonic function defined over the interval $[x_0, x_c]$, then $t = g(x) = f^{-1}(x)$, and so g is therefore a concave, monotonic function with negative curvature. We consider applying a single perturbation $\Delta x$ to the system, drawn from an even probability distribution $P(\Delta x)$, with vanishing tails at a point $x_p$. We choose $x_p$ so that $x_c - x_p$ and $x_p - x_0$ are both large compared to $\Delta x$: that is we are not close to the boundary of the predictable state already. From the

evenness of $P$, then the impact of the perturbation on the variable $x$, in expectation, is zero:

$$\langle \Delta x \rangle = \langle X - x_p \rangle = 0 \tag{3}$$

However this is not in general the case for $\langle \Delta t \rangle$:

$$\langle \Delta t \rangle = \langle g(x_p + \Delta x) - g(x_p) \rangle = \langle G_p(\Delta x) \rangle \tag{4}$$

where $G_p$, as a translation of $g$ is also concave. Using the definition of the expectation we find:

$$\langle \Delta t \rangle = \int\limits_{-\infty}^{\infty} G(\Delta x) \cdot P(\Delta x) d\Delta x \tag{5}$$


where the limits are justified by our requirement for vanishing tails for $P$ and our choice of $x_p$.

Because $P(\Delta x)$ is even, then if we Taylor expand $G$ we will lose all odd powers on integration and be left with:

$$\langle \Delta t \rangle = \int\limits_{-\infty}^{\infty} \sum_{n=1}^{\infty} \frac{1}{(2n)!} \frac{d^{2n}G}{d(\Delta X)^{2n}} \cdot P(\Delta x) d\Delta x \tag{6}$$

And having taken $\Delta X$ to be small, we can drop high order terms:

$$\langle \Delta t \rangle = \int\limits_{-\infty}^{\infty} \frac{1}{2} \frac{d^2G}{d(\Delta X)^2} \cdot P(\Delta x) d\Delta x \tag{7}$$


Because $G$ is concave, this derivative is negative and we obtain

$$\langle \Delta t \rangle = -\int\limits_{-\infty}^{\infty} \frac{1}{2} \left| \frac{d^2G}{d(\Delta X)^2} \right| \cdot P(\Delta x) d\Delta x = -I \tag{8}$$

Where I is positive definite. Therefore any convex perturbation growth will be delayed (i.e. intermittency persisted) by small symmetric random perturbations, *and the average lifetime of the stable regime will increase*. The case for a negatively oriented



variable with concave curvature is of course precisely equivalent. For the special case of $f(t) = x_0 e^{\lambda t}$, and $P(\Delta x) = \mathcal{N}(0, \sigma)$
we obtain analytically $\langle \Delta t \rangle = \frac{-\sigma^2}{2\lambda x_p^2}$ (see Appendix B). We see that larger $\sigma$ corresponds to longer persistence (remembering
our overall weak noise approximation), as does small $\lambda$, indicating that nearly stable intermittent dynamics can be most sub-
stantially persisted. We also see in this case that smaller $x_p$ leads to larger drift, indicating that perturbations soon after the
system has entered the intermittent state are most impactful. While we've considered only a single perturbation, the reasoning

generalises to a continuous stochastic process, where now the $\langle \Delta t \rangle$ term we have calculated is a spatially dependent drift term.
Our argument is similar in many ways to the consideration of a Poincaré recurrence or first-return map taken along the unstable
manifold, where the stochastic variable plays the role of the chaotic mapping.

    This mechanism is very general as it based solely on geometric arguments, and indeed the crux of the argument can be
understood solely from examination of figure 8. Any low dimensional, convexly growing instability in the climate system

should be susceptible to the same stochastic suppression. We can also see why only the blocking regime showed dramatic
persistence increases, as it is the only regime where the variables evolve with a constant sign of curvature, and where there is
only one unstable dimension.

## 4   Conclusions

In this paper we have investigated the phenomenon of stochastically-reinforced regime persistence in a six-mode truncated

model of a barotropic $\beta$-plane channel. We have shown that the changes in regime lifetime observed by Kwasniok (2014)
are associated with shifts in the system's power spectra, and have obtained an understanding of the dynamical origins of this
counter-intuitive behaviour.

    We observe that the model contains a vacillation between chaotic and ordered flow states, with a predominately turbulent
zonal flow, divided by transitions into a quasi-stationary blocked state. Similarities can be seen to the behaviour of the 1-

D Pomeau-Maneville map which demonstrates the same intermittently chaotic dynamics. Similar observations of movements
between chaos and order in atmospheric systems have been recently documented in Shen et al. (2021), and the dynamics we see
in our simple six-equation model bear remarkable similarities to the conclusions of Faranda et al. (2016), which applies cutting-
edge dynamical systems metrics to observational data and concludes that persistent atmospheric blocking in the Atlantic is due
to the influence of an unstable fixed point in a single basin chaotic attractor of prevailing zonal flow. However, the overall role

of such order-to-disorder transitions have in the atmosphere is still an open question.

    We have introduced a simple geometric argument with broad general applicability, which explains how a zero-mean stochas-
tic perturbation will act to delay the growth of one-dimensional instabilities and so can prolong intermittent periods of order in
a chaotic dynamical system.

    This stochastic persistence is intensified, and a heavy tail introduced into the lifetime of blocking events, through chance

excursions to the neighbourhood of a deterministically inaccessible fixed point, where the overall deterministic tendency is
negligible. Here, the dynamics are almost entirely noise-dominated, and the system will remain close to the fixed point until
it scatters some critical distance away, where the deterministic tendency starts to become significant again. We have discussed



how this scenario connects us to the theory of hitting time problems, where it can be proven that the distribution of times taken for Brownian motion to leave a radius $a$ of an initial point is fat tailed.

Ultimately, if stochastic forcing is increased past $\sigma$=0.008, the blocking regime starts to become less persistent. This is not clearly linked to a resonance effect as suggested in K14 and C04, but is instead a result of entering a completely noise-dominated dynamics. This effect is of little relevance for practical applications of stochastic parameterisation to models, as the stochastic terms will almost always be second-order compared to deterministic tendencies. However, even for $\sigma < 0.008$ there is a considerable sensitivity of regime statistics to the exact amplitude of the stochastic forcing. In the context of complex

models, this suggests attention should be given to the amplitude of stochastic perturbations, just as is commonly done for deterministic parameters. If these results qualitatively generalise, then a stochastic forcing that is too weak may not stabilise slowly varying modes of the earth-system to the degree seen in observations.

It remains to be seen however how relevant these mechanisms are in more complex atmospheric models, and indeed the impact of stochastic physics on climate model regime structure are still unclear, complicated by high sampling variability in

the Euro-Atlantic region Dor (2022). Order-to-disorder transitions played an important role in the stochastic persistence of the CdV system, and much of the direct relevance of these results therefore depends on whether such transitions have a significant presence in the true atmospheric flow – a currently unanswered question. A natural next step is to ask how stochasticity impacts regimes in quasigeostrophic settings, and to explore how the mechanisms uncovered here evolve (or indeed, are rendered extinct) as the complexity of the flow increases. In order to better link simple model behaviour to that of complex models we

must continue to develop tools that can be used to study both kinds of systems.

## Appendix A:  Derivation of the Charney deVore 1979 model

Here we derived the CdV79 model introduced in section 2 from the shallow water equations, marking simplifying approximations in **boldface**. The shallow water system is a simple but highly useful model of fluid flow. Consider a **single layer** of fluid, with **constant density** $\rho$, **negligible viscosity**, a vertical pressure gradient deriving solely from gravity (i.e. the fluid is

in **hydrostatic equilibrium**), and which, as the name implies, has a representative **horizontal length scale** $L$ **much larger than the vertical length scale** $H$. The dynamics of this system are completely determined by the momentum equation and the mass-conservation equation.

The momentum equation in a rotating frame is given by

$$\frac{\partial \mathbf{v}}{\partial t} + (\mathbf{v} \cdot \boldsymbol{\nabla})\mathbf{v} + \mathbf{f} \times \mathbf{v} = -g\boldsymbol{\nabla} H \tag{A1}$$

while the mass-conservation equation is

$$\frac{\mathrm{D}\,H}{\mathrm{D}\,t} + H\boldsymbol{\nabla} \cdot \mathbf{v} = 0 \tag{A2}$$



Where $\mathbf{v}$ is velocity, $\mathbf{f}$ represents Coriolis forces, $g$ is simply the gravitational constant (**assumed constant**) and $H$ is the depth of the fluid column. Here the mass-conservation simply expresses that if velocities converge at a point, the height of the fluid column must rise in response. Application of the identity $(\mathbf{a} \cdot \mathbf{b})\mathbf{c} = -\mathbf{a} \times (\mathbf{b} \times \mathbf{c}) + (\mathbf{a} \cdot \mathbf{c})\mathbf{b}$ to equation A1 gives us

$$\frac{\partial \mathbf{v}}{\partial t} + [(\boldsymbol{\nabla} \times \mathbf{v}) \times \mathbf{v} + \frac{1}{2}\boldsymbol{\nabla} v^2] + \mathbf{f} \times \mathbf{v} = -g\boldsymbol{\nabla} H \tag{A3}$$

and by introducing the absolute vorticity $\xi = \boldsymbol{\nabla} \times \mathbf{v}$, and taking the curl of both sides, we get the shallow water vorticity equation:

$$\frac{\partial \boldsymbol{\xi}}{\partial t} + \boldsymbol{\nabla} \times ((\boldsymbol{\xi} + \mathbf{f}) \times \mathbf{v}) = 0 \tag{A4}$$

Again, we apply the triple product identity and the chain rule and find:

$$\frac{\partial \boldsymbol{\xi}}{\partial t} + (\mathbf{v} \cdot \boldsymbol{\nabla})(\boldsymbol{\xi} + \mathbf{f}) + (\boldsymbol{\xi} + \mathbf{f})(\boldsymbol{\nabla} \cdot \mathbf{v}) = 0 \tag{A5}$$

Where $\boldsymbol{\nabla} \cdot \boldsymbol{\xi} = \boldsymbol{\nabla} \cdot (\boldsymbol{\nabla} \times \mathbf{v}) = 0$ by definition and $(\boldsymbol{\xi} \cdot \nabla)\mathbf{v} = 0$ because $\boldsymbol{\xi}$, as a curl of horizontal wind, does not project onto the variability of $\mathbf{v}$. Indeed, at this point we note that both $\boldsymbol{\xi}$ and $\mathbf{f}$ have no horizontal components, and so we focus only on the scalar equation for the z-component, letting $\xi := \xi_z$ and $f := f_z$ for convenience. The divergent velocity term can be rewritten with the aid of equation A2 by rearranging:

$$\boldsymbol{\nabla} \cdot \mathbf{v} = \frac{-1}{H}\frac{\mathrm{D}\,H}{\mathrm{D}\,t} \tag{A6}$$

and substituting into equation A5. This leaves us with

$$\frac{\mathrm{D}\,(\xi + f)}{\mathrm{D}\,t} - \frac{(\xi + f)}{H}\frac{\mathrm{D}\,H}{\mathrm{D}\,t} = 0 \tag{A7}$$

which can be put into a clearer form using the chain rule:

$$\frac{\mathrm{D}}{\mathrm{D}\,t}\left[\frac{(\xi + f)}{H}\right] = 0 \tag{A8}$$

That is, the potential vorticity $q = \frac{\xi + f}{H}$ is materially conserved. If the thickness of the fluid increases, then the absolute vorticity must likewise increase, due to the stretching of the flow.

We divide our fluid height $H$ into some average fluid depth $D$ and an anomaly component $h(x,y)$ describing the height of surface orography, so that $H = D - h(x,y)$. We assume **shallow orography** so we may take the limit where $\frac{h}{D} << 1$ and Taylor expand $H$, truncating at first order so that $\frac{1}{H} \approx \frac{1}{D} + \frac{h}{D^2}$:

$$\frac{\mathrm{D}}{\mathrm{D}\,t}\left[\xi + f + \frac{(\xi + f)h(x,y)}{D}\right] = 0 \tag{A9}$$



In the midlatitude atmosphere this is a reasonable approximation at the largest scales. For example, if we let $D$ be the height of the tropopause – around 8 km – and we let $h$ be the average elevation of Europe or North America – between 500 and 800 metres – then the limit holds. We also choose to place ourselves in the **quasigeostrophic limit** of low Rossby number, where $\xi < f$. Finally, we take a $\beta$ **plane approximation** to the Coriolis force, and this brings us to the barotropic vorticity equation in the presence of shallow orography:

$$\frac{\mathrm{D}}{\mathrm{D}t}\left[\xi + \beta y + \frac{f_0}{D}h(x,y)\right] = 0 = \frac{\mathrm{D}q}{\mathrm{D}t} \tag{A10}$$

We define $\gamma = \frac{f_0}{D}$ and introduce the streamfunction $\Psi$, given by $u = -\frac{\partial \Psi}{\partial y}$, $v = \frac{\partial \Psi}{\partial x}$, $\xi = \nabla^2 \Psi$. We also generalise slightly by introducing an external source (or sink) of vorticity $F$. Writing our material derivative out explicitly, we now have:

$$\frac{\partial \nabla^2 \Psi}{\partial t} - \frac{\partial \Psi}{\partial y}\cdot\frac{\partial}{\partial x}[\nabla^2\Psi + \gamma h + \beta y] + \frac{\partial \Psi}{\partial x}\cdot\frac{\partial}{\partial y}[\nabla^2\Psi + \gamma h + \beta y] = F \tag{A11}$$

Or more concisely

$$\frac{\partial \nabla^2 \Psi}{\partial t} + \mathcal{J}(\Psi, \nabla^2\Psi + \gamma h) - \beta\frac{\partial \Psi}{\partial x} = F \tag{A12}$$

Where the Jacobian operator is given by

$$J(A,B) := \frac{\partial A}{\partial x}\frac{\partial B}{\partial y} - \frac{\partial B}{\partial x}\frac{\partial A}{\partial y} \tag{A13}$$

Taking our **forcing to be a linear relaxation** towards a background state, we now have our explicit equation for the evolution of $\Psi$:

$$\frac{\partial}{\partial t}\boldsymbol{\nabla}^2\Psi = -J(\Psi, \boldsymbol{\nabla}^2\Psi + \gamma h) - \beta\frac{\partial \Psi}{\partial x} - C(\Psi - \Psi^*) \tag{A14}$$

This is the CdV79 equation, before spectral truncation, where $\Psi^*$ is our background streamfunction. We consider the flow in a **channel geometry**, with periodic boundary conditions in the zonal direction, and with no-penetration and no-slip conditions in the meridional direction. Of course the real atmosphere has no hard meridional boundaries; however the impacts of the subtropical and midlatitude jets have been shown to produce a waveguiding effect that meridionally confines low-frequency variability in a similar way, both in low-order models Hoskins and Ambrizzi (1993); Yang et al. (1997) and in observations Branstator (2002), which makes this a reasonable first approximation.

We consider a channel of dimensionless zonal length $x = [0, 2\pi]$, and meridional length $y = [0, b\pi]$, so that $b$ then is a free parameter controlling the shape of the channel.





## A1   Discretising the model

To convert this infinite dimensional partial differential equation into a system of ordinary differential equations, we can use a Galerkin projection. This involves expanding our spatially dependent variables ($\Psi$, $\Psi^*$, and $h$) in terms of an infinite linear sum of complete spatial basis functions, labelled $\{\Phi(x,y)_i\}$:

$$\Psi(x,y,t) = \sum_i^\infty \Psi_i(t)\Phi_i(x,y) \tag{A15}$$

$$\Psi^*(x,y) = \sum_i^\infty \Psi_i^*\Phi_i(x,y) \tag{A16}$$

$$h(x,y) = \sum_i^\infty h_i\Phi_i(x,y) \tag{A17}$$

We start by defining this basis abstractly, requiring the basis functions to be eigenfunctions of the Laplacian operator and the zonal derivative:

$$\nabla^2\Phi_i = -a_i^2\Phi_i \tag{A18}$$

$$\frac{\partial\Phi}{\partial x} = b_i\Phi_i \tag{A19}$$

The Jacobian term is the thorniest point in the spectral analysis, as it non-linearly mixes spectral modes together. It is the impact of this term that prevents an arbitrary spectral truncation of equation A14 being a solution of the full equation. In full generality, we simply consider that the Jacobian of each pair of basis functions has some projection on every other basis function:

$$J(\Phi_j,\Phi_k) = c_{ijk}\Phi_i \tag{A20}$$

The projection coefficient of the jth and kth modes on the ith mode, $c_{ijk}$, is computable simply through a direct pointwise product of the ith mode and the Jacobian, integrated over the domain and appropriately normalised:

$$c_{ijk} = \frac{\int_0^{2\pi}\int_0^{b\pi}\Phi_i J(\Phi_j,\Phi_k)dydx}{\int_0^{2\pi}\int_0^{b\pi}dydx} \tag{A21}$$





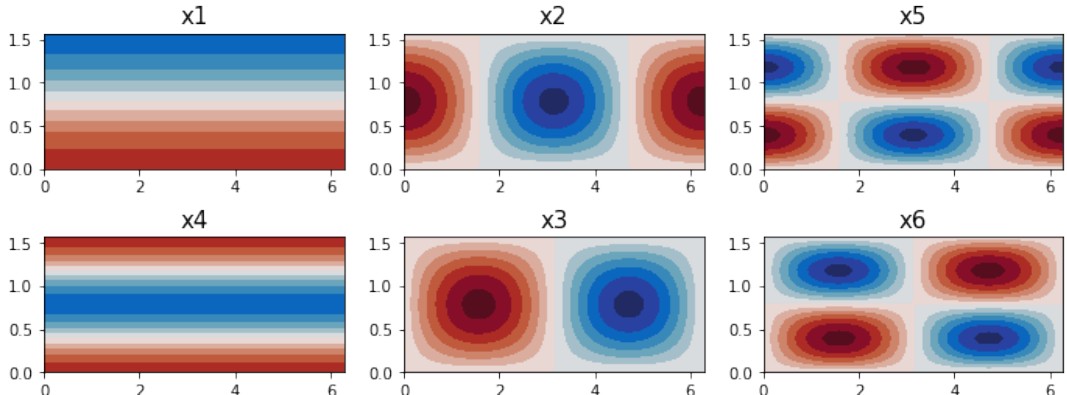

**Figure A1.** The real-valued basis functions used in the spectrally truncated CdV79 system, plotted as streamfunctions

Under these constraints, we can now rewrite equation A14 projected onto the ith basis function as:

<a name="325"></a>

$$\frac{\partial}{\partial t}\Psi_i = \beta a_i^{-2} b_i \Psi_i - C(\Psi_i - \Psi_i^*) - \sum_{j,k} a_i^{-2} c_{ijk}(a_k^2 \Psi_j \Psi_k - \gamma \Psi_j h_k) \tag{A22}$$

Where the double sum indexes over all basis functions. We make an explicit choice of basis functions now, taking a Fourier basis defined as:

$$\begin{cases} \sqrt{2}\cos\left(\frac{ny}{b}\right) & \text{if } m = 0 \\ \sqrt{2}e^{imx}\sin\left(\frac{ny}{b}\right) & \text{else} \end{cases}$$

where the meridional mode $n > 0$ and the zonal mode $m$ can take any non-integer value. It is easy to verify that these

basis functions satisfy the boundary conditions and eigenfunction constraints defined above. To actually solve these spectrally transformed equations, it will be necessary to introduce a truncation, disregarding wave modes exceeding a cutoff wavenumber. Here we choose to introduce a zonal threshold of $|m| \leq 1$ and a meridional threshold of $n \leq 2$, **restricting us to just 6 basis functions**, as in CdV79, C04 and K14.

Although up to this point our approximations have been well motivated physically, we cannot rigorously justify introducing

such a drastic truncation without resorting to self-interest: such a system is much easier to study, and can be understood in more detail than a high-dimensional model. Nonetheless, we may reason that any phenomena observed in this truncated model, which will have to derive solely from the dynamics of the largest scales, may have analogues in the true atmospheric circulation, although there they will be shaped and influenced by coupling to smaller scales. The resulting basis functions are shown in figure A1.





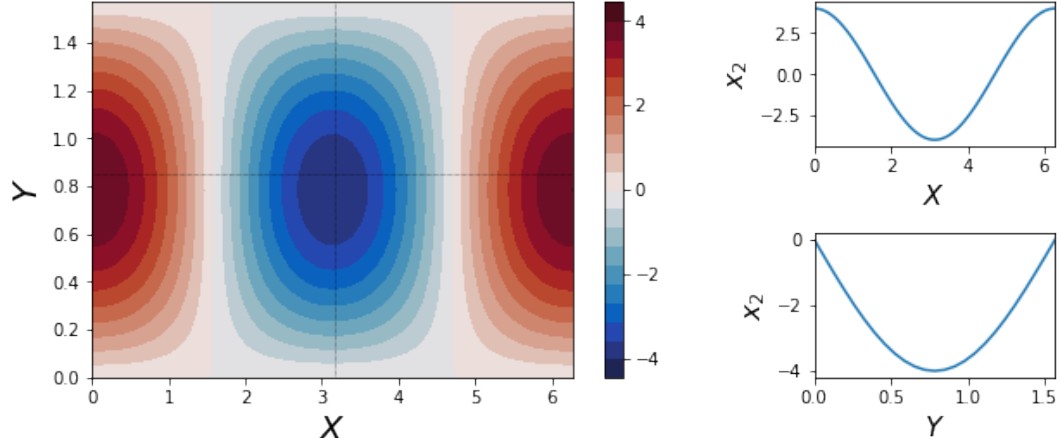

**Figure A2.** The orographic profile used in the CdV79 system, with a trough in the centre of the domain and a ridge over the periodic boundary. Sub-panels show cross sections across the domain, following the dotted black lines.

Concretely, while the true spectral equation for the largest wave modes, equation A22, contains an infinite number of nonlinear coupling terms, after truncation we are reduced to only 36 such terms, representing only interactions between the resolved modes: the ability for small scales to feed vorticity into the untruncated large scales is neglected.

     We choose our **linear relaxation to be zonally symmetric**, representing the forcing effect of the pole-to-equator temperature gradient. We choose our orography to include a single peak straddling the periodic boundary, with a trough in the centre of the domain. $h(x,y) = \gamma\sqrt{2}\cos(x)\sin\left(\frac{y}{b}\right)$ (this is shown in figure A2).

Obtaining the coefficients in A22 is now simply a matter of calculating integrals, but is very involved, and is best performed using symbolic computation software (an implementation using sympy (Meurer et al., 2017) can be found at https://github.com/joshdorrington/Thesis_materials/blob/main/10d_mode_derivation.ipynb). After computing coefficients, we now diverge slightly from CdV79, transforming our variables to a real basis, as in C04, given by:

$$x_1 = \tfrac{1}{b}\Phi_{0,1} \qquad\qquad x_4 = \tfrac{1}{b}\Phi_{0,2}$$
$$x_2 = \tfrac{1}{\sqrt{2}b}(\Phi_{1,1} + \Phi_{-1,1}) \quad x_3 = \tfrac{i}{\sqrt{2}b}(\Phi_{1,1} - \Phi_{-1,1})$$
$$x_5 = \tfrac{1}{\sqrt{2}b}(\Phi_{1,2} + \Phi_{-1,2}) \quad x_6 = \tfrac{i}{\sqrt{2}b}(\Phi_{1,2} - \Phi_{-1,2})$$

     This transformation is purely cosmetic; the behaviour of the system is unchanged. At the end of this process we obtain an explicit six-equation system:



$$\dot{x}_1 = -C(x_1 - x_1^*) + \tilde{\gamma}_1 x_3$$

$$\dot{x}_2 = -Cx_2 + \beta_1 x_3 - \alpha_1 x_1 x_3 - \delta_1 x_4 x_6$$

$$\dot{x}_3 = -Cx_3 - \beta_1 x_2 + \alpha_1 x_1 x_2 + \delta_1 x_4 x_5 - \gamma_1 x_1$$

$$\dot{x}_4 = -C(x_4 - x_4^*) + \epsilon(x_2 x_6 - x_3 x_5) + \tilde{\gamma}_2 x_6$$

$$\dot{x}_5 = -Cx_5 + \beta_2 x_6 - \alpha_2 x_1 x_6 - \delta_2 x_4 x_3$$

$$\dot{x}_6 = -Cx_6 - \beta_2 x_5 + \alpha_2 x_1 x_5 + \delta_2 x_4 x_2 - \gamma_2 x_4$$

(A23)

Where colours encode the linear relaxation, Coriolis forces, orographic impacts and non-linear advection. We see that there

are far less than the 36 promised non-linear terms per equation: most quadratic interactions integrated over the domain vanish due to symmetry considerations. $\{\alpha_n\}, \{\beta_n\}, \{\delta_n\}, \{\gamma_n\}, \{\tilde{\gamma_n}\}$ and $\epsilon$ are simple geometric constants of $\mathcal{O}(1)$ given in C04.

There are now a number of dimensionless free parameters in the model, which we set equal to the values used in K14 and C04:

– $x_i^*$: The forcing vector, representing thermal relaxation towards a background state. We use a zonally symmetric forcing
profile with $x_1 = 0.95, x_4 = -0.76095$, all other terms zero (shown in figure A3).

– $C$: The thermal relaxation timescale. Set to $C = 0.1$, describing a damping time of $\sim$10 days.

– $\gamma$: The orographic amplitude. Set to $\gamma = 0.2$, corresponding to a 200-metre amplitude.

– $\beta$: The Coriolis parameter. Set to $\beta = 1.25$, defining a central latitude of $45°$.

– $b$: Defines the channel half-width. Set to $b = 0.5$, which gives a channel of  6300 km x 1600 km





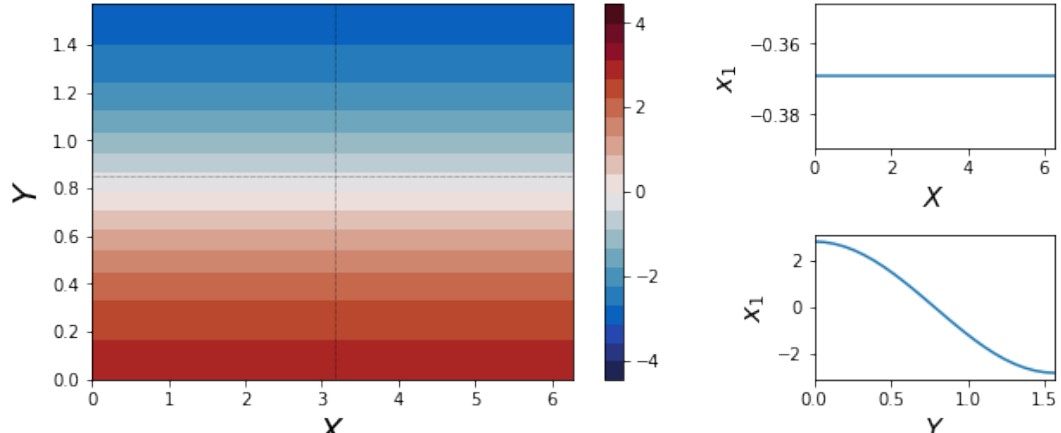

**Figure A3.** The background state in the Crommelin '04 formulation of the CdV system. It is zonally symmetric and represents the impact of the meridional temperature gradient. Again, sub-panels show cross sections across the domain, following the dotted black lines.

## Appendix B:  Impact of a Gaussian perturbation on exponential growth

In section 3 we found that

$$\langle \Delta t \rangle = -\int_{-\infty}^{\infty} \frac{1}{2} \left| \frac{d^2 G}{d(\Delta X)^2} \right| \cdot P(\Delta x) d\Delta x = -I \tag{B1}$$

If we consider $x$ to be undergoing exponential growth, and subject it to a Gaussian perturbation so that $x = f(t) = x_0 e^{\lambda t}$, and $P(\Delta x) = \mathcal{N}(0, \sigma)$, then we can obtain an analytical form for $\Delta t$. Remembering that $G_p(\Delta x) = g(X) - g(x_p)$, and seeing that $g(x) = \frac{1}{\lambda} \log \left( \frac{x}{x_0} \right)$, we have:

$$\langle \Delta t \rangle = \int_{x_0}^{x_c} \frac{1}{\lambda} \log \left( \frac{X}{x_p} \right) \cdot \frac{1}{\sigma \sqrt{2\pi}} e^{-\frac{1}{2} \left( \frac{X - x_p}{\sigma} \right)^2} dX \tag{B2}$$

We transform our variables to $w = \frac{\Delta x}{\sigma} = \frac{X - x_p}{\sigma}$, which implies $\frac{X}{x_p} = \frac{x_p + \sigma w}{x_p} = 1 + \frac{\sigma}{x_p} w$ and $dX = \sigma \cdot dw$. Our limits of integration become $w_0 = \frac{x_0 - x_p}{\sigma} < 0$ and $w_c = \frac{x_c - x_p}{\sigma} > 0$.

If, as in the main text, we assume $\sigma$ is small relative to a typical length scale of $x$ (that is the perturbation is vanishingly unlikely to force the system into the chaotic regime directly) then these limits will be very large in absolute magnitude. Given this and the exponentially decaying tails of the Gaussian distribution, we approximate $w_0 \approx -\infty$, $w_c \approx +\infty$. This gives us:

$$\langle \Delta t \rangle = \frac{1}{\lambda \sqrt{2\pi}} \int_{-\infty}^{+\infty} \log \left( 1 + \frac{\sigma}{x_p} w \right) \cdot e^{-\frac{w^2}{2}} dw \tag{B3}$$



While not directly integrable, we can make progress by Taylor expanding the logarithm. Firstly we recognise that as the Gaussian distribution is even in $w$, then all odd terms in the Taylor expansion will vanish. Secondly we note that $\frac{\sigma}{x_p} w = \frac{\Delta x}{x_p}$

will be small if, as above, $\sigma$ is small, so we can drop terms of 4th order and higher in $w$. Thus we end up with:

$$\langle \Delta t \rangle = \frac{-\sigma^2}{2\lambda x_p^2 \sqrt{2\pi}} \int\limits_{-\infty}^{+\infty} w^2 \cdot e^{-\frac{w^2}{2}} dw \qquad (B4)$$

Which, from any book of identities, we can find gives:

$$\langle \Delta t \rangle = \frac{-\sigma^2}{2\lambda x_p^2 \sqrt{2\pi}} \cdot \sqrt{2\pi} = \frac{-\sigma^2}{2\lambda x_p^2} \qquad (B5)$$

*Author contributions.* JD performed all simulations and drafted the manuscript, under the supervision of TP

*Acknowledgements.* The authors would like to thank Glenn Shutts, Daan Crommelin and Frank Kwasniok for enlightening discussions.





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
