# Peer review of "On the interaction of stochastic forcing and regime dynamics"

_EGUsphere, 2022_

## Referee Comment (RC2)

[referee-annotated manuscript omitted]

---

## Author Response (AR1)

**Response to reviewer comments**

We would like to thank both reviewers for their close reading and helpful comments. A significant restructuring of our introduction, mechanisms and conclusions has taken place to include the literature brought to our attention by reviewer 2, and we have replaced figure 5 with a different presentation of the power spectra which more clearly shows the non-monotonic evolution of low-frequency variability with stochastic forcing.

**Reviewer 1**

**The quality of the figures is very poor, with some labels and numbers barely readable.**

We apologise for the poor quality of some figures, which we have now corrected.

**I would test the peaks in the PSDs in Figure 5 against red noise surrogates to be sure that the noise generates new long-time behaviour in the system (resonances). It is done for instance in Groth, A., & Ghil, M. (2015). Monte Carlo Singular Spectrum Analysis (SSA)**

Considering also a comment by the second reviewer on this figure, figure 5 has been totally redesigned. We have not implemented the suggested red noise test, but instead we now show in panel b) an average of the low-frequency power as a function of stochastic noise amplitude. This is because our intention with this figure is not to compare the dynamics against any particular reference spectrum, but to demonstrate its non-monotonic evolution as stochastic forcing increases. We believe this new version of the figure better meets this aim.

**\* The presentation of the proposed mechanism for the delay of the regime on page 9 and 10 is poor.**

An attempt to clarify the notation in fact led to notation errors in a previous draft. We have fixed these.

**For instance:**

    **- capital letters are mixed with lower-case ones.**

Fixed.

    **- you should display the axis in Figure 8.**

Done.

    **- the symbol dt and dx in Figure 8 must be defined, they do not relate to anything in the text.**

Now changed to Δt and Δx as in the text.

**- line 178: you say that P has vanishing tails at x_p and I don't understand. To me you choose P with vanishing support outside [x0-xp, xc - xp] (I would rather say with support on [x0-xp, xc - xp], but alright).**

A misplaced comma was to blame for the confusion. We have also now changed the double negative.

**- line 179: you state something about Δx, but it doesn't mean anything, you have to express that in term of the moments of P.**

Reworded to be clearer:

"Informally, this amounts to choosing a point $x_p$ that is far from the edge of the predictable regime and so there is only a vanishing probability of $\Delta x$ directly triggering a regime transition. When the standard deviation of $P$ is small compared to $x_c-x_0$, this will apply to most values of $x$ in the interval."

**- line 193: You say that I is positive definite. I am not a native english speaker but to me this expression is reserved for matrices and bilinear forms. Please check.**

Now we simply say "I is positive".

**- In the case of a concave curvature (for a positively oriented variable), would the persistence be decreased? Could you comment on that?**

We now comment on this:

"while for a positively oriented variable with concave curvature or a negatively oriented variable with convex curvature decreased persistence would be seen. For the linear case with zero curvature, there is of course no impact of the perturbation on persistence."
* * *
**I have also a more general comment. The view that regimes can be described by fixed points has been criticized and is somehow a bit outdated. The authors cite Faranda et. al. (2016) as a sort of "proof" that blocking is involving an unstable fixed point. However, you have to note that this study was involving only one field (z500), while blocking is most likely a multi-dimensional problem. Also, to my knowledge, they collapsed toghether all blockings from all seasons in the northern hemisphere to perform their**

**analysis, while it is well known that blocking is very different in winter and in summer. Finally, this study presents as a sort of argument of authority that their method using the concept of extremal index detects only unstable fixed points, while if you read the papers they are citing, you discover that this is valid only for idealized uniformly expanding systems, and that it can detect also periodic orbits. In that sense, I would take the claim that blocking is uniquely defined by unstable fixed points with a grain of salt. Reality is probably more complicated, with developed chaos playing probably a role. I would be surprised if the structure involved is as simple as a fixed point.**

In response to comments by the other reviewer, calling to our attention a sizable body of related literature on stochastically forced transient chaos in maps, we have substantially altered our introductions and conclusions. As a result of the length limitations of NPG, this discussion of Faranda et al. has been cut. We agree that a fixed point perspective in the real atmosphere is likely simplistic, and this is still discussed in our introduction. We now refer more to the dynamics of intermittent repellors, and to concepts of transient chaos. The important topic of UPO dynamics has naturally arisen in this context. We now discuss UPOs on lines 232-236

**Minor corrections**

 **- lines 126-127: The expression "by the time" here seems a bit odd. Please check.**

Removed as part of rewrites.

 **- line 167: "As the stochastic becomes". What does it mean? Please check.**
Now reads: "As the stochastic forcing becomes"

 **- line 240: Citation should be (Dor, 2022) ? Also does not appear correctly in the references list.**
Should be Dorrington 2022: this is now fixed.

**Reviewer 2**

**The authors of the paper study a simplified version of a barotropic model of atmospheric circulation and find that increasing the intensity of additive noise (upto a point) makes the persistence of regimes stronger, especially that of the blocked state, aligning with the same observation by Kwasniok (2014) and others. They then develop a more generic argument and derivation, arriving at a formula for the expected trajectory life time (in a regime) as a function of noise intensity. In the weak noise limit, they find a quadratic enhancement effect. Unfortunately, this is all known and derived long ago, including the quadratic formula. A good starting point is Sec.**

**4.1.2 "Enhancement of Transient Lifetime by Noise" of the book 'Transient chaos' by Lai and Tel (2011):**

We thank the author for his review and for bringing these important prior results to our attention. We have taken time to survey this new (to us) literature in some detail, and have restructured our introduction, presentation of mechanisms, and conclusions heavily to include it properly.

Upon careful consideration of the derivation in Lai and Tell 2011, and in the three papers by Riemann cited within, we think our result goes slightly beyond a rederivation. While the 'flavour' of the approach and the resulting lifetime formula are very similar, the underlying deterministic functions are very different. Riemann's result is based on concave, non-invertible maps, whereas our result relies on a (positively oriented) convex and monotonic tendency equation. Nevertheless we now give proper precedence to these results, and integrate the perspective they provide throughout our discussion.

Including these results has added a new strand to the paper: highlighting the idea of considering circulation regime dynamics in terms of transient chaos, and connecting regime transitions to escape rates. Certainly in terms of understanding the impacts of stochastic forcing this is a valuable addition.

**Minor comments identified from annotated pdf**

**Did they really claim that these would be unpredictable? Or was it actually unpredictable back then? Was there weather forecast service at that time?**

Yes, weather forecast services have been running since Robert Fitzroy established the Met Office in 1854, and in fact it was Charney who developed the first operational numerical weather prediction model which went live in 1955! Nevertheless, they do not comment explicitly on the predictability of the regime transitions in the paper so we have taken that word out.

**Then why do you write in the beginning of the sentence "reproduction of regime dynamics"?**

Changed to be clearer: "While the reproduction of regime dynamics in barotropic annulus experiments \citep{weeks1997} suggested that regimes could occasionally be associated with fixed points even in untruncated flows…"

**This does not sound to be related to "stabilise regimes" that you discuss in the previous sentence**

Changed to be clearer: "...by increasing persistence…"

**Relative to what? As per Fig. 3b, you can have 3 +ve LE's. Is it?? [in relation to quasiperiodicity]**

Different definitions may be used in different fields, but we intend quasiperiodic to mean there are clear peaks in the power spectrum indicating preferred frequency bands. It is in this sense also that we describe the dynamics as weakly chaotic, in that the behaviour is somehow close to a regular oscillation..

**What is the justification for this? (Additive forcing)**

This was done primarily to follow on from the approach of Kwasniok 2014. Sensitivity tests with red noise forcing showed no qualitatively distinct impact.

**At this point, it wouldn't hurt to remind the reader that sigma is the additive noise intensity. [for sigma !=0…]**

Done.

**I'm not sure if we are seeing the regime of blocking here. Is it dark blue and blanked out, or light blue?**

Dark blue is plotted on top of light blue so there is no blanking out here. The blocking state is associated with 1+ve eigenvalue, but clearly some other regions of phase space, assigned to the transitional regime, are as well.

**How do you determine regimes when the noise intensity is different? Is it meaningful to stick with the same "algorithm" (?), or, do you adjust it somehow?**

Figure 4 and its discussion provide justification for this choice. By training a separate HMM for each sigma value, we found qualitatively similar assignments and dynamics, at least for the range of noise amplitudes up until sigma=0.016. Post-hoc the smooth variations in regime lifetime distributions help solidify our confidence in this approach.

**What threshold? Plot a graph of power in a low freq band vs sigma. Is there a jump in this curve?**

This was a poor choice of language. However we have now replaced figure 5 with an alternate plot, following your suggestion of plotting band averaged power, as we feel this is clearer.